# Molecular Pharmacology of *Gelsemium* Alkaloids on Inhibitory Receptors

**DOI:** 10.3390/ijms25063390

**Published:** 2024-03-16

**Authors:** Ana M. Marileo, César O. Lara, Anggelo Sazo, Omayra V. Contreras, Gabriel González, Patricio A. Castro, Luis G. Aguayo, Gustavo Moraga-Cid, Jorge Fuentealba, Carlos F. Burgos, Gonzalo E. Yévenes

**Affiliations:** 1Department of Physiology, Faculty of Biological Sciences, Universidad de Concepcion, Concepción 4070386, Chile; 2Millennium Nucleus for the Study of Pain (MiNuSPain), Santiago 8331150, Chile

**Keywords:** *Gelsemium* alkaloids, glycine receptor, GABA_A_ receptor, electrophysiology, bioinformatics

## Abstract

Indole alkaloids are the main bioactive molecules of the *Gelsemium* genus plants. Diverse reports have shown the beneficial actions of *Gelsemium* alkaloids on the pathological states of the central nervous system (CNS). Nevertheless, *Gelsemium* alkaloids are toxic for mammals. To date, the molecular targets underlying the biological actions of *Gelsemium* alkaloids at the CNS remain poorly defined. Functional studies have determined that gelsemine is a modulator of glycine receptors (GlyRs) and GABA_A_ receptors (GABA_A_Rs), which are ligand-gated ion channels of the CNS. The molecular and physicochemical determinants involved in the interactions between *Gelsemium* alkaloids and these channels are still undefined. We used electrophysiological recordings and bioinformatic approaches to determine the pharmacological profile and the molecular interactions between koumine, gelsemine, gelsevirine, and humantenmine and these ion channels. GlyRs composed of α1 subunits were inhibited by koumine and gelsevirine (IC_50_ of 31.5 ± 1.7 and 40.6 ± 8.2 μM, respectively), while humantenmine did not display any detectable activity. The examination of GlyRs composed of α2 and α3 subunits showed similar results. Likewise, GABA_A_Rs were inhibited by koumine and were insensitive to humantenmine. Further assays with chimeric and mutated GlyRs showed that the extracellular domain and residues within the orthosteric site were critical for the alkaloid effects, while the pharmacophore modeling revealed the physicochemical features of the alkaloids for the functional modulation. Our study provides novel information about the molecular determinants and functional actions of four major *Gelsemium* indole alkaloids on inhibitory receptors, expanding our knowledge regarding the interaction of these types of compounds with protein targets of the CNS.

## 1. Introduction

The *Gelsemium* genus of flowering plants belongs to the Loganiaceae family and comprises five North American, East Asian, and Chinese native species. Extracts of these plants have traditionally been employed in Asian folk medicine to treat various illnesses, such as neuralgia, sciatica, rheumatoid arthritis, and pain [1,2]. The earliest records of this kind of plant’s usage date back to the *The Shennong Emperor’s Classic of Materia Medica* (up to the early third century A.D.), which detail their therapeutic properties and toxicity [1,2]. Diverse *Gelsemium* species have been used to treat pathological conditions. For example, *Gelsemium elegans* has traditionally been used to treat eczema,traumatic injuries, pretibial ulcers and myiasis, and has also been used as an analgesic to relieve sciatica and rheumatoid arthritis, while *Gelsemium sempervirens* has been employed to treat cancer, spinal inflammation, and back pain, and as an antispasmodic [1,2].

Several studies have characterized the phytochemical profile of *Gelsemium* plants [2,3,4]. These reports describe the species as a rich source of natural compounds, including iridoids, coumarins, steroids, and alkaloids. Indole alkaloids have been characterized as the main active molecules of the *Gelsemium* species. The four principal compounds are gelsemine, koumine, gelsevirine, and humantenmine (also known as gelsenicine).

Studies using purified indole alkaloids have shown many biological effects in both in vitro and in vivo assays [2,3,4]. These actions range from antitumor activity to antioxidant and anti-inflammatory effects [2,4,5]. Additional reports have shown the positive actions of several of these alkaloids in pathological state models of the central nervous system (CNS), including anxiety [6,7], persistent pain [8,9,10,11,12], and Alzheimer’s disease [13]. Nevertheless, *Gelsemium* indole alkaloids are intrinsically toxic to animals and humans. The toxicity symptomatology profile, which frequently includes asphyxia, dyspnea, convulsions, and respiratory arrest, is consistent with unfavorable effects at the CNS [1,2,3,4]. These findings suggest that a part of both the beneficial and toxic actions of indole *Gelsemium* alkaloids is associated with the modulation of molecular targets involved in the control of neuronal activity.

To date, molecular targets underlying the *Gelsemium* indole alkaloids’ biological actions at the mammalian CNS remain unclear. Although several membrane receptors and enzymes are involved in the mechanisms underlying *Gelsemium* alkaloids’ beneficial actions, evidence of the direct modulatory actions of these compounds on specific biological targets or the characterization of the molecular determinants involved in protein–alkaloid interactions is mainly lacking.

Electrophysiological studies have determined that gelsemine is a functional modulator of glycine receptors (GlyRs) and type A GABA receptors (GABA_A_Rs) [14,15], which are the main ligand-gated ion channels controlling CNS synaptic inhibition [16]. Gelsemine exerts subunit-specific actions on GlyRs composed of α subunits. Previous studies reported that gelsemine displayed a bell-shape modulation on currents through homomeric α1GlyRs and a concentration-dependent inhibition on α2 and α3GlyRs [14]. Spinal GlyRs were also inhibited by the alkaloid and showed an IC_50_ of about 42 μM [14]. On the other hand, gelsemine inhibited recombinant and native GABA_A_Rs and showed IC_50_ values of about 55–75 μM [15]. Experimental evidence from radioligand assays and electrophysiological analyses suggest that gelsemine actions on these ion channels occurs in a competitive manner. For instance, using GlyRs from spinal cord tissue, Zhang and coworkers found that gelsemine displaces ^3^H-strychnine-binding curves to the right, calculating an IC_50_ gelsemine value on native GlyRs of about 40 μM [10,12]. Electrophysiological studies performed on recombinant GlyRs found that gelsemine displaces glycine concentration response curves of homopentameric α1 to the left, and α2 and α3 GlyRs curves to the right [14]. Similarly, Marileo and colleagues observed that gelsemine displaces the GABA concentration response curve to the right [15], which is consistent with competitive inhibition. These findings have provided support to other lines of research suggesting the GlyRs’ or GABA_A_Rs’ participation in the mechanisms related with gelsemine analgesic and anxiolytic actions [6,10,11,12]. However, the molecular sites involved in the interaction between *Gelsemium* alkaloids, and these ion channels are still undefined. Furthermore, it is currently unknown whether indole alkaloids other than gelsemine exert functional actions on these receptors. Thus, a compelling assessment of *Gelsemium* alkaloids’ pharmacological potential and toxicological relevance is limited by our poor understanding of the molecular mechanisms underlying their therapeutic and toxic actions. Therefore, we aimed to provide a comprehensive view of the molecular interactions between the four major indole alkaloids (i.e., koumine, gelsemine, gelsevirine, and humantenmine) with these ion channels.

## 2. Results

We first examined the sensitivity of the most abundant GlyR α subunit of the CNS, the α1 subunit [16], to koumine, gelsevirine, and humantenmine (Figure 1A–D). The application of koumine and gelsevirine inhibited the glycinergic currents of α1GlyRs from a concentration of 10 μM and showed no potentiation (Figure 1A,B). The alkaloid-mediated inhibition followed a sigmoidal fit with IC_50_ values of 31.5 ± 1.7 μM (n = 6) for koumine and 40.6 ± 8.2 μM (n = 9) for gelsevirine (Table 1). Koumine sensitivity significantly differed for α1 and α2GlyRs (Table 1). In addition, the gelsevirine modulation showed significant differences in n_H_ values (Table 1). Similar experiments showed that humantenmine did not significantly modulate α1GlyRs in a concentration range of 10 μM to 300 μM (Figure 1A,B). No inhibition was obtained with 300 μM of the alkaloid (−19.2 ± 7.8%, n = 6) (Figure 1A,B). These data suggest that indole alkaloids have different modulatory profiles on homomeric α1GlyRs.

We examined the heteromeric α1βGlyRs modulation to assess the β subunits’ influence on the alkaloid-mediated modulation. This GlyR configuration is expressed at glycinergic synapses and is vital for inhibition in the spinal cord [16]. Koumine and gelsevirine displayed comparable inhibitory actions on these receptors, showing similar percentages of maximal modulation (Figure 1C,D and Table 1). Koumine and gelsevirine also inhibited the heteromeric α2β and α3β GlyRs function (Figure 1B–D and Table 1). Koumine and gelsevirine (Figure 1B–D and Table 1) also inhibited the heteromeric α2β and α3β GlyRs function. Contrary to homomeric GlyRs, koumine sensitivity and nH values for gelsevirine inhibition did not significantly differ in heteromeric GlyRs, suggesting that α subunits’ integration to the pentamer may regulate the alkaloid actions. Humantenmine could not modify the heteromeric GlyRs function examined (Figure 1D).

Then, we examined whether these alkaloids’ modulatory profile on GlyRs was preserved on GABA_A_Rs. As previously shown, gelsemine can also inhibit GABA_A_Rs’ function but with a significantly lower potency and efficacy than GlyRs [14,15]. Koumine sensitivity of recombinant α1β2γ2 GABA_A_Rs, the most widely expressed GABA_A_R subtype in the mammalian brain [16], revealed -34.0 ± 5.3% of inhibition (n = 4), which was significantly lower than the koumine-induced inhibition of GlyRs (Figure 1E,F). Further recordings showed that humantenmine did not elicit any detectable alterations on the GABA-evoked currents (Figure 1E,F). Along with previous reports [14,15], these results show that koumine, gelsevirine, and gelsemine exert inhibitory actions on the GlyR and GABA_A_R function, whereas the alkaloid humantenmine was inactive. Moreover, these data indicate that GlyRs are more sensitive to *Gelsemium* alkaloid actions than GABA_A_Rs.

The functional results previously described suggest that gelsemine, koumine, and gelsevirine have common physicochemical features that match with acceptor sites on inhibitory channels, while the chemical structure of humantenmine possibly lacks critical requirements to stably bind and modulate these membrane proteins. We performed molecular docking assays using the structures available for GlyRs and GABA_A_Rs to start the molecular examination of the alkaloid’s interaction with these ion channels [17,18,19,20,21,22,23]. Due to their relevance as the structural domains responsible for binding agonists, antagonists, and allosteric modulators [24], the docking procedures focused on the extracellular domain (ECD) and transmembrane domains (TMD). Our bioinformatic assays revealed that a major percentage of the alkaloid–GlyR (≈81–95%) and alkaloid–GABA_A_R (≈74%) complexes were located on the receptor ECD, while few interactions were positioned on the TMDs (Appendix A). At the ECD, the alkaloids displayed favorable interactions with the orthosteric site, which correspond to the glycine or GABA-binding site. Next, we executed in silico extra precision docking score measurements on the orthosteric site of homomeric α1, α2, and α3 subunits to obtain putative interaction insights between the alkaloids with different GlyR subunits (Figure 2A,B). In these analyses, we included the classical GlyR inhibitor, strychnine, as a reference compound binding to the orthosteric site [17,18]. Docking score values are predicted binding affinities (in kcal/mol) for the molecule’s capacity to interact with a defined binding site. These in silico assays were used to calculate the feasible ligand–protein interactions of a given site within a protein structure. A docking score was computed for each ligand–receptor interaction, referred to as a binding pose. The most negative value indicates a more favorable binding energy, thus suggesting a more stable ligand–receptor complex. To provide a broader view of potential ligand–receptor interactions, we used box plots (percentiles 25 and 75 and median) alongside maximum and minimum docking score values (indicated by whiskers) to comprehensively describe the full docking score dataset. Gelsemine, koumine, and gelsevirine interaction with the α GlyR subunits’ orthosteric site showed similar docking scores, comparable to the values displayed by strychnine (Figure 2B). On the other hand, putative interactions between humantenmine and the orthosteric site exhibited docking scores that were shifted to less negative values (Figure 2B). This molecular interactions profile was replicated on GABA_A_Rs composed of α1β2γ2 (Figure 2C,D). Gelsemine, koumine, and bicuculline (a classical GABA_A_Rs antagonist) interacted with the orthosteric site located in the interface between the α and β subunits. At the same time, humantenmine showed a docking score distribution that was shifted to less negative values (Figure 2D). Further interaction analyses between the alkaloids and the α-β binding interface of heteromeric GlyRs displayed favorable interactions (Appendix A). Conversely, the interface composed of GABA_A_R α1 and γ2 subunits could not bind any of the alkaloids studied. Altogether, these results correlate well with previous reports [10,12,14,15] and suggest a leading role of the orthosteric site as being responsible for binding the alkaloid to GlyRs and GABA_A_Rs. However, additional binding sites are still possible (see Appendix A), especially considering the presence of the subunit-specific effects displayed by gelsemine on α1 GlyRs versus other GlyR conformations [14].

To functionally examine the ECD relevance as a main determinant of the GlyR alkaloid modulation and the subunit-specific actions of gelsemine on α1GlyRs, we studied chimeric receptors where the ECD was exchanged between the α1 and α2 subunits (Figure 3A) [25]. Previous studies showed that α1GlyRs were potentiated by 10–50 μM gelsemine, while α2GlyRs were inhibited by these alkaloid concentrations [14]. The functional relevance of the ECD was assessed by testing the subunit-specific potentiation elicited by gelsemine on these receptors (Figure 3B,C). Our control experiments on wild-type receptors demonstrated that 10 μM of gelsemine potentiated α1GlyRs, while 50 μM of the alkaloid inhibited α2GlyRs. The exchange of the α2 ECD with its α1 counterpart generated a receptor potentiated by gelsemine, similar to α1GlyRs (i.e., chimeric receptor α1α2, Figure 3B,C). Likewise, the exchange of the α1 ECD with its α2 counterpart (i.e., chimeric receptor α2α1) displayed an inhibitory effect with gelsemine, like α2GlyRs (Figure 3B,C). These results suggest that both the binding of gelsemine and its subunit-specific effects are exclusively related to the ECD of GlyRs, ruling out the involvement of other alkaloid binding sites in other ion channel domains.

Considering our functional data and evidence coming from diverse groups [10,12,14,15], GlyR orthosteric site appears to be the critical molecular site for the alkaloid–receptor interaction. To confirm this idea, we examined the gelsemine, koumine, gelsevirine, and humantenmine molecular interactions with residues within the GlyR orthosteric site. Due to subunit-specific effects, we centered these analyses on α1GlyRs. Molecular docking showed that residues of two adjacent subunits collectively participate in the alkaloid binding. Diverse residues from the complementary subunit (i.e., chain A) (F44, F63, L64, R65, S129, L127) and the main subunit (i.e., chain B) (S158, F159, G160, Y202, F207) contribute to stabilizing the interaction among gelsemine, koumine, and gelsevirine with α1GlyRs, tested at pH = 7.0 (Figure 4A). Interestingly, the interaction of charged nitrogen groups on these alkaloids with residues on GlyRs contributes to their stability. In both koumine and gelseverine, a cation–pi interaction was observed between the charged nitrogen and phenylalanine residues (F44, F63, and F159) (Figure 4A). Gelsemine exhibited a hydrogen bond between the charged nitrogen group and a serine residue (S129) (Figure 4A). On the other hand, humantenmine could anchor into the orthosteric site but did not display detectable interactions with any residues (Figure 4A). Further sequence analyses showed that these residues were fully conserved in α2 and α3GlyRs, suggesting a similar contribution to the binding of alkaloids (Appendix A). We performed electrophysiological recordings on α1GlyRs containing mutations on several of the amino acids identified to experimentally corroborate these in silico findings. Mutations on the orthosteric site may create nonfunctional receptors, complicating alkaloid modulation analyses. The mutagenesis plan was designed by first assessing in silico how substitutions may affect glycine binding. Our simulations revealed that F63A and G160E mutations could significantly reduce alkaloid binding while preserving a proportion of glycine binding. Consequently, these mutated α1GlyRs were synthesized and studied by electrophysiology (Figure 4B–D). Our electrophysiological studies indicated that F63A and G160E mutated α1GlyRs are receptors with altered glycine sensitivity, in agreement with previous reports [26,27]. Nevertheless, after 2–3 days post transfection, the cells displayed stable currents at glycine concentrations of 1–2 mM, allowing for the examination of the *Gelsemium* alkaloid sensitivity. The glycine-evoked currents through F63A and G160E mutated α1GlyRs were insensitive to gelsemine concentrations triggering potentiation (i.e., 10 μM) or inhibition (200 μM) (Figure 4B–D). Further recordings showed that koumine and gelsevirine could also not exert detectable effects on these mutated GlyRs (Figure 4D). The F63A showed a complete loss of functional strychnine modulation, which correlates with previous electrophysiological findings [26]. In contrast, the G160E mutation still retained a proportion of the strychnine inhibition of the glycine-evoked currents (wild-type = −98.5 ± 5.1% (n = 5); F63A = −1.7 ± 3.1% (n = 4); G160E, −48.5 ± 13.9% (n = 6), 2 μM strychnine. ANOVA followed by Tukey post hoc test. Differences were significant. F(2, 11) = 5.81: *, *p* < 0.05, wild-type versus G160E; ***, *p* < 0.001, wild-type versus F63A; *, *p* < 0.05, F63A versus G160E). We explored the physicochemical requirements the alkaloids may fulfill to exert a functional modulation on these receptors through pharmacophore modeling to have a complete vision of the alkaloid–receptor interaction (Figure 4E). These analyses showed that the main requirements for a functional action of these alkaloids are three hydrophobic groups, an aromatic ring, and a positively charged nitrogen group outside the indole group. These requirements are fully satisfied by gelsemine, koumine, gelsevirine, and strychnine. Humantenmine only fulfilled two out of three requirements and did not show a positively charged nitrogen group. Collectively, these data provide an integral view to explain the presence or absence of the GlyR functional modulation by the main *Gelsemium* alkaloids.

## 3. Discussion

A growing pool of evidence has demonstrated biological actions mediated by *Gelsemium* alkaloids. Most of the research has focused on the most abundant *Gelsemium* indole alkaloids: gelsemine, gelsevirine, koumine, and humantenmine. These alkaloids displayed biological activities against diverse pathological states and robust toxic actions in mammals [1,2,3,4]. Despite phytochemical and pharmacological relevance, the biological protein targets underlying the effects of *Gelsemium* alkaloids remain unclear. In particular, functional and biochemical information describing the interactions of these alkaloids with protein targets is virtually lacking. Using electrophysiological recordings combined with molecular modeling and site-directed mutagenesis, we described the molecular determinants involved in the functional modulation of inhibitory receptors by the most prominent *Gelsemium* alkaloids.

Based on their chemical structures, the indole-type alkaloids are classified in six groups: gelsemine, koumine, humantenine, gelsedine, sarpagine, and yohimbine [1,2,3,4]. Our functional data indicated that the most prominent alkaloids of the gelsemine and koumine groups were active on inhibitory channels, whereas humantenmine, a representative compound of the gelsedine type, was largely inactive. In general terms, the indole *Gelsemium* alkaloid subtypes possess either oxindole or indole cores in combination with diverse chemical entities. Our pharmacophore analyses suggest that the indole groups contribute to providing an aromatic ring and a hydrophobic core, which are requirements for functional activity. The chemical groups accompanying the indole groups should contribute two additional hydrophobic groups and a positively charged nitrogen acting as a hydrogen bond donor. These requirements matched well with our functional activity profile and were also consistent with the physicochemical features of strychnine, a GlyR reference competitive alkaloid. These data support the idea that introducing or subtracting discrete chemical substituents within the accompanying groups of the indole cores may switch the properties of a given alkaloid in terms of its functional actions on these channels. Therefore, it will be interesting to describe the structure–activity relationships of different *Gelsemium* alkaloids’ groups to identify novel inhibitory channel modulators of natural origin. This concept is highlighted by the biphasic modulation exerted by gelsemine on α1GlyRs. Future studies combining functional assays with in silico screenings may contribute to generating a compelling profile of the *Gelsemium* indole alkaloids’ actions on inhibitory ion channels and in other types of receptors.

Despite our study being restricted to the cellular and molecular level, we believe our findings contribute, at least in part, to better interpreting *Gelsemium* alkaloids’ actions in vivo. Gelsemine and koumine are the most studied *Gelsemium* alkaloids regarding their beneficial effects in pathological models. The literature suggests that both alkaloids share key features, such as analgesic actions [8,9,10,11,12] and anxiolytic effects [6,7]. Some of these properties have also been investigated for gelsevirine, showing comparable actions with gelsemine and koumine [6,28]. Although the proposed mechanisms underlying these effects have been diverse, the direct binding and activation of GlyRs by these alkaloids have been postulated as a key event [10]. The participation of GlyRs has been supported by results showing that the application of strychnine diminished the actions of *Gelsemium* alkaloids [6,10,11,12]. In addition, gelsemine and koumine binding to GlyRs was demonstrated using competitive displacement assays [10,12]. Nevertheless, previous reports and our results consistently show that *Gelsemium* alkaloids are mostly of GlyRs and GABA_A_Rs’ antagonists [14,15]. Thus, a direct GlyR or GABA_A_R activation on the mechanisms underlying, for instance, the analgesic or anxiolytic effects of *Gelsemium* alkaloids, should be taken with care. On the other hand, our results with humantenmine correlate well with the observations reported by Liu and coworkers [6], as they showed that this alkaloid did not display actions on anxiety models. However, it should be noted that the alkaloid doses and concentrations used to investigate beneficial effects were, generally, lower than those required to obtain a robust GlyRs or GABA_A_Rs modulation [7,10,11,12]. Overall, we think that additional studies are needed to formulate a more definitive relationship between the GlyR or GABA_A_R modulation by *Gelsemium* alkaloids and their beneficial actions in paradigms of pathological states.

On the other hand, we believe that the data presented here, and previous reports [14,15], help provide a rational neurophysiological framework to explain the toxicity elicited by gelsemine, koumine, and gelsevirine. Experimental evidence indicates that these three alkaloids are competitive antagonists of GlyRs and GABA_A_Rs. These results establish a common mechanism of action for the *Gelsemium* alkaloids and traditional antagonists of these types of receptors: strychnine, picrotoxin, and bicuculline [16]. Those similarities imply that gelsemine, koumine, and gelsevirine will decrease the glycinergic and GABAergic function, generating a loss of inhibitory control in the CNS which matches with a major part of the symptomatology of the *Gelsemium* intoxication [1,2,3,4]. In contrast, our functional results with humantenmine suggest that its toxicity is unrelated to the modulation of inhibitory channels. In vivo studies have shown that humantenmine is the most potent *Gelsemium* indole alkaloid in terms of its toxicology, with LD_50_ values lower than 0.2 mg/kg [4]. In similar assays, gelsemine and koumine displayed LD_50_ values higher than 50 mg/kg, consistent with a different mechanism of action [4]. Our electrophysiological results and previous reports [14,15] show that alkaloids’ concentrations that decrease around 25–50% of the currents through GlyRs and GABA_A_Rs are generally higher than 50 µM, a value nearly equivalent to 16 μg/mL. The maximal modulation concentrations, which in most cases generate an 80–95% decrease in the currents, are reached with 100–300 µM, nearly the equivalent of 30–95 µg/mL. Interestingly, these concentration ranges correlate well with lethal plasma concentrations of *Gelsemium* alkaloids reported in humans, which are in the range of 25–50 µg/mL for koumine and 13–30 µg/mL for gelsemine [29]. It should also be noted that pharmacokinetic studies performed with 11 *Gelsemium* alkaloids described gelsemine as the only compound detectable in brain tissue after 3 days of systemic application, while koumine was fully depleted after 1 day [30]. Based on these reports, it is possible to suggest that at least a part of the toxicity exerted by the gelsemine-type and koumine-type indole compounds relates to the GlyRs or GABA_A_Rs inhibition, while other targets mediate the gelsedine-type alkaloids’ toxic actions. Thus, it will be relevant continuing the study on additional CNS targets for the toxic actions of these alkaloids, especially considering that *Gelsemium* preparations are still used in humans and recent cases of intoxication and food contamination have been reported [4,31,32]. We believe that these studies will help to develop safe and more targeted antidotes against *Gelsemium* poisoning.

Altogether, our results outline the molecular features involved in the modulation of the inhibitory receptor by the main *Gelsemium* indole alkaloids. Since our studies indicate that subtle differences in the alkaloid structures determine the functional modulation of inhibitory receptors, we suggest that future research focused on the functional and structural mapping of diverse *Gelsemium* indole alkaloids interactions with multiple membrane ion channels and receptors may provide a compelling view to understand their biological actions on the mammalian CNS.

## 4. Materials and Methods

### 4.1. Chemicals

Gelsemine, koumine, gelsevirine, and humantenmine (>99% purity) were obtained from ChemFaces (Wuhan, China). All other chemicals were purchased from Tocris (Bristol, UK), Hello-Bio (Bristol, UK), Sigma-Aldrich (St. Louis, MO, USA), or AK Scientific (Union City, CA, USA).

### 4.2. Cell Culture, Plasmids, and Transfection

HEK293 cells (CRL-1573; American Type Culture Collection, Manassas, VA, USA) were cultured using standard procedures [14,15]. The cells were transfected with plasmids encoding the following proteins: (i). rat GlyR α (1, 2, 3) subunits alone or combined with rat β subunits (Uniprot Accession Numbers: Q546L7, P22771, P20236, P20781); and (ii). rat GABA_A_R α1 combined with rat β2 and γ2 subunits (Uniprot Accession Numbers: P62813, P63138, P18508). The EGFP expression was used as a marker of successful protein expression. The transfection was performed using Lipofectamine 2000 (Invitrogen, ThermoFisher, Carlsbad, CA, USA) using the fabricant protocol. The plasmids used to express GlyRs and GABA_A_Rs have been previously described [14,15,25]. To track the successful expression of GlyR β subunits and the GABA_A_ γ2 or β2 subunit, these proteins were subcloned in the pIRES2-EGFP vector. The transfection ratio to obtain β-containing GlyRs was 1:5 of α:β, while the expression of γ2-containing GABA_A_ receptors was 1:2:5 of α:β:γ2 [14,15]. Chimeric GlyRs were described in a previous report [25]. Site directed mutagenesis was performed by Charm Gene Science Mutagenex LLC. (Cumming, GA, USA). All the constructs were checked by full-length sequencing (SNPSaurus, Eugene, OR, USA).

### 4.3. Electrophysiology

Glycine or GABA-evoked currents were recorded from transfected HEK293 at room temperature (20–24 °C) as previously described [14,15,25]. In brief, patch electrodes (3–4 MΩ) were pulled from borosilicate glasses in a PC-100 puller (Narishige, Tokyo, Japan); then, they were filled with an intracellular solution, which contained (in mM) 120 CsCl, 8 EGTA, 10 HEPES (pH 7.4), 4 MgCl_2_, 0.5 GTP, and 2 ATP. The extracellular solution contained (in mM) 140 NaCl, 5.4 KCl, 2.0 CaCl_2_, 1.0 MgCl_2_, 10 HEPES (pH 7.4), and 10 glucose. Whole-cell patch-clamp recordings were performed at −60 mV with an Axoclamp 200B (Molecular Devices, Sunnyvale, CA, USA) or with HEKA EPC-10 (HEKA Elektronik GmbH, Reutlingen, Germany) amplifiers. The data acquisition was executed by using Clampex 10.1 or PatchMaster v2.11 software. Data analysis was accomplished off-line using Clampfit 10.7 (Axon Instruments, Sunnyvale, CA, USA). Exogenous glycine or GABA-evoked currents were obtained using a gravity-based perfusion device. Brief (3–5 s) pulses of agonist together with alkaloids were applied to cells. Stock solutions of the alkaloids were prepared in high purity distilled water. Aliquots of these stock solutions were diluted in reservoirs containing extracellular solution. The effects of the alkaloids on glycine or GABA-evoked currents were obtained using a co-application of sub-saturating agonist concentration (EC_10–20_) together with the alkaloid. The EC_50_ for wild-type recombinant GlyRs was 76 ± 4 μM (α1, n = 5), 91 ± 6 μM (α2, n = 5), 150 ± 10 μM (α3, n = 4), 71 ± 4 μM (α1β, n = 6), 105 ± 9 μM (α2β, n = 4), and 243 ± 13 μM (α3β, n = 5). The EC_50_ for α1β2γ2 GABA_A_R was 5.0 ± 0.3 μM (n = 6). The EC_50_ for chimeric GlyRs was 55 ± 3 μM (α1α2, n = 7) and 104 ± 9 μM (α2α1, n = 6). The mutated α1GlyRs did not show current saturation even when using 10–15 mM of glycine [26,27]. Consequently, the EC_50_ values were estimated to be higher than 8 mM for F63A and higher than 3 mM for G160E. Concentrations higher than 10 mM of glycine were difficult to test due to seal variabilities. The modulation percentage was calculated using the following equation: Percentage change = 100 × ((I_alkaloid_ − I_agonist_)/ I_agonist_), where I_alkaloid_ is the current in the presence of a given concentration of alkaloid, and I_agonist_ is the amplitude of the control glycine or GABA current elicited by the activation of a given subunit combination. The concentration–response curve parameters for the alkaloid inhibition (IC_50_, n_H_, and the maximal current inhibition) were obtained from the fitted curve of normalized concentration–response data points to the equation I_agonist_ = I_max_ (agonist)n_H_/((agonist)n_H_ + (EC_50_)n_H_). The IC_50_ curves are best-fit mean ± SE from pooled data. I_agonist_ is the current in the presence of a given sub-saturating concentration of GABA, n_H_ is the Hill coefficient, EC_50_ is the concentration required for half-maximal response, and I_max_ is the maximum amplitude of the current. The inhibition constant (K_i_) was calculated using the Cheng–Prusoff equation (K_i_ = IC_50_/((2 + ([Agonist]/EC_50_)^n^_H_)^1/n^_H_ − 1). The concentration of agonist corresponds to the sub-saturating agonist concentration used to obtain the IC_50_ for each alkaloid on a given GlyR subtype.

### 4.4. Molecular Docking and Bioinformatic Procedures

Protein–ligand docking was performed using the structures of α1β, α2β, α3 GlyRs, and α1β2γ2 GABAAR obtained from the Protein Data Bank (PDB ID: 7TU9, 7KUY, 5CFB, 6X3S) [17,18,19,20,21,22,23]. All structures bounded either strychnine or bicuculline, indicating a closed conformational state location. Before docking simulations, all protein structures were prepared with Maestro’s v2020-3 Protein Preparation Workflow tool software. This process included the addition of hydrogens, H-bond assignments optimization, the protonation states determination at pH 7 ± 0.2, and filling in missing side chains with Prime. Similarly, the gelsemine, koumine, gelsevirine, and humantenmine structures were retrieved from the PubChem database (CID: 5390854, 44583834, 14217344, 158212) and prepared using LigPrep to generate ionization states at pH 7 ± 0.2 and possible conformations for each alkaloid (Schrödinger, LLC, New York, NY, USA, 2020).

All site-directed docking calculations were performed using Glide (Schrödinger, LLC, New York, NY, USA, 2020) with a grid centered on the orthosteric binding site of the α/α, α/β interfaces on GlyR and α/β, α/γ interfaces on GABA_A_R. Predictions were made employing the extra-precision (XP) configuration with a post-docking minimization that included 10 poses per ligand, from which the best pose was selected to represent each protein–ligand complex. Analysis of the complexes encompassed the structural and energetic parameters summarized in the docking score values. In silico mutagenesis was conducted using the Residue and Loop Mutation module from Maestro. After substituting the selected amino acid, refinement was performed through implicit solvent minimization, including all residues within 5 Å around the mutation. Pharmacophore modeling was carried out using Phase (Schrödinger, LLC, New York, NY, USA, 2020) in standard configuration, with several features for each hypothesis ranging from 5 to 6, and a threshold of 60% for active molecules.

### 4.5. Statistical Analyses

All results are presented as mean ± SEM. Statistical analysis and graph plotting were performed with Origin (version 6.0 or 8.0). Values of *p* < 0.05, *p* < 0.01, and *p* < 0.001 were considered statistically different. Statistical comparisons were performed using paired or unpaired Student’s *t*-tests. Multiple comparisons were analyzed with ANOVA followed by a Tukey post hoc test.

## Figures and Tables

**Figure 1 ijms-25-03390-f001:**
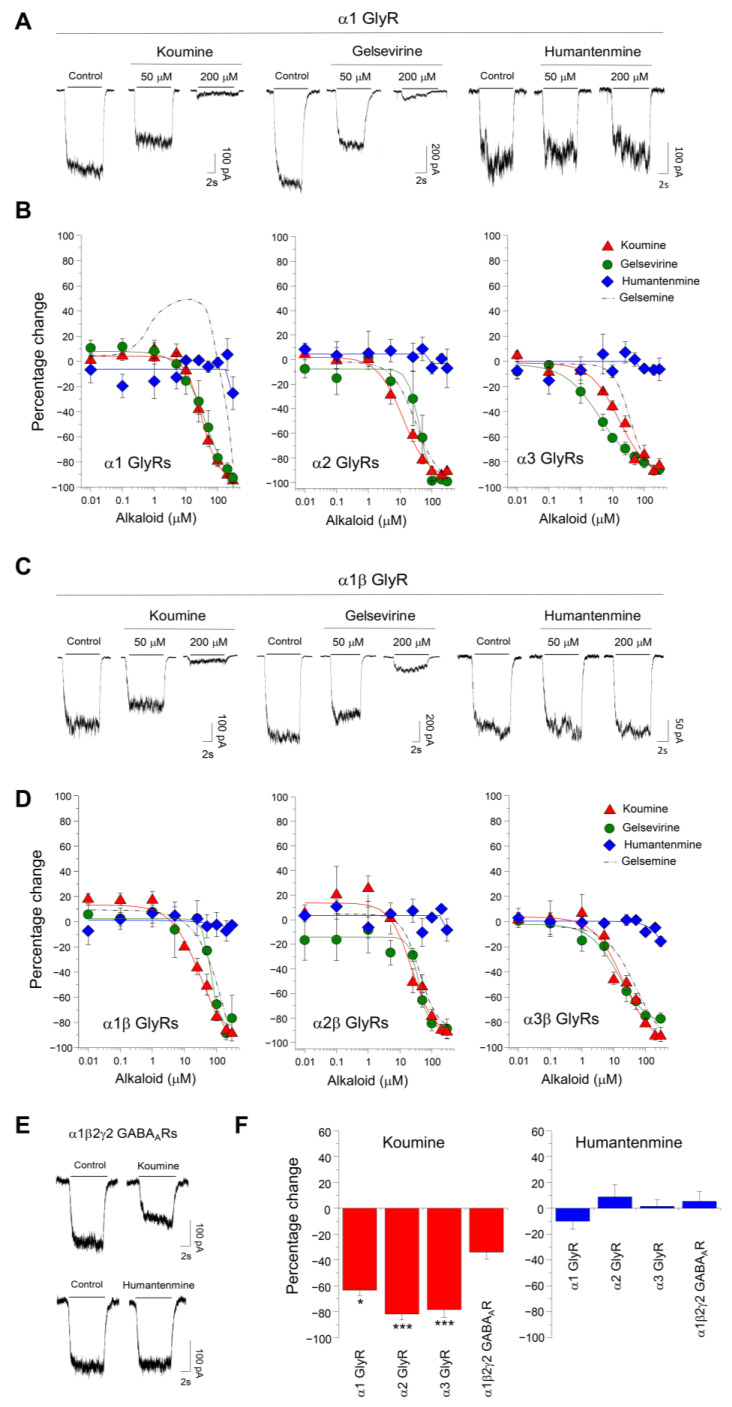
Modulation of recombinant GlyRs and GABA_A_Rs by *Gelsemium* alkaloids. (**A**) Current traces before and during the application of koumine, gelsevirine, or humantenmine to cells expressing α1GlyRs. (**B**) Concentration response curves (0.01–300 μM) of alkaloids on homomeric α1, α2, and α3 GlyRs. Currents were evoked using 35 μM (α1), 60 μM (α2), or 65 μM (α3) of the agonist glycine. The dashed lines describe the gelsemine sensitivity [14]. (**C**) Current traces before and during the application of koumine, gelsevirine, or humantenmine to cells expressing α1β GlyRs. (**D**) Concentration response curves (0.01–300 μM) of alkaloids on α1β, α2β, and α3β GlyRs. The currents were evoked using 30 μM (α1β), 60 μM (α2β), or 70 μM (α3β) of glycine. The dashed lines describe the gelsemine sensitivity [14]. (**E**) Current traces before and during the application of koumine or humantenmine (50 μM) to cells expressing α1β2γ2 GABA_A_Rs. (**F**) The graph summarizes the sensitivity of GABA-evoked currents to 50 μM of koumine or humantenmine. Currents were evoked using 1 μM of GABA. *, *p* < 0.05, koumine-induced inhibition of α1β2γ2 GABA_A_Rs versus α1GlyRs; *** *p* < 0.001; koumine-induced inhibition of α1β2γ2 GABA_A_Rs versus α2 and α3GlyRs. ANOVA followed by Tukey post hoc test, F(3, 17) = 0.2916. Data are presented as means ± SEM.

**Figure 2 ijms-25-03390-f002:**
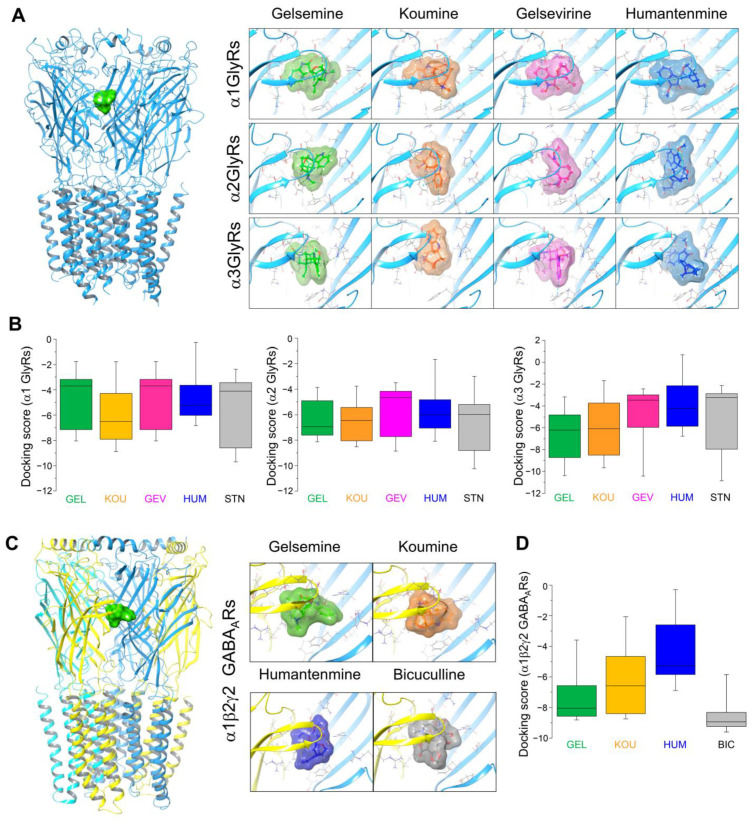
Putative binding sites of *Gelsemium* alkaloids within the orthosteric sites of GlyRs and GABA_A_Rs. (**A**) The left panel shows gelsemine binding to homopentameric α1GlyRs. Panels on the right show an enhanced view of the predicted binding of gelsemine, koumine, gelsevirine, andhumantenmine to the orthosteric sites of α1, α2, and α3GlyRs. Glycine binding is shown in Appendix A. (**B**) The boxed graphs summarize the docking scores for the gelsemine (GEL), koumine (KOU), gelsevirine (GEV), humantenmine (HUM), and strychnine (STN) interaction to the orthosteric sites. (**C**) The left panel shows the binding of gelsemine to α1β2γ2 GABA_A_Rs. Panels on the right show an augmented vision of the putative binding of gelsemine, koumine, humantenmine, and bicuculline to the α1β2γ2 GABA_A_R orthosteric site. (**D**) The graph shows the docking score values for the interaction of gelsemine (GEL), koumine (KOU), humantenmine (HUM), and bicuculline (BIC) to the orthosteric site. The boxed graphs show medians (middle line) and quartile ranges (25–75, box borders). Whiskers indicate the maximal and the minimal docking score values. The parameters of strychnine and bicuculline are also shown as reference compounds. The number of binding conformations for each alkaloid were as follows: gelsemine (13), koumine (30), gelseverine (19), humantenmine (62), strychnine (37), and bicuculline (16).

**Figure 3 ijms-25-03390-f003:**
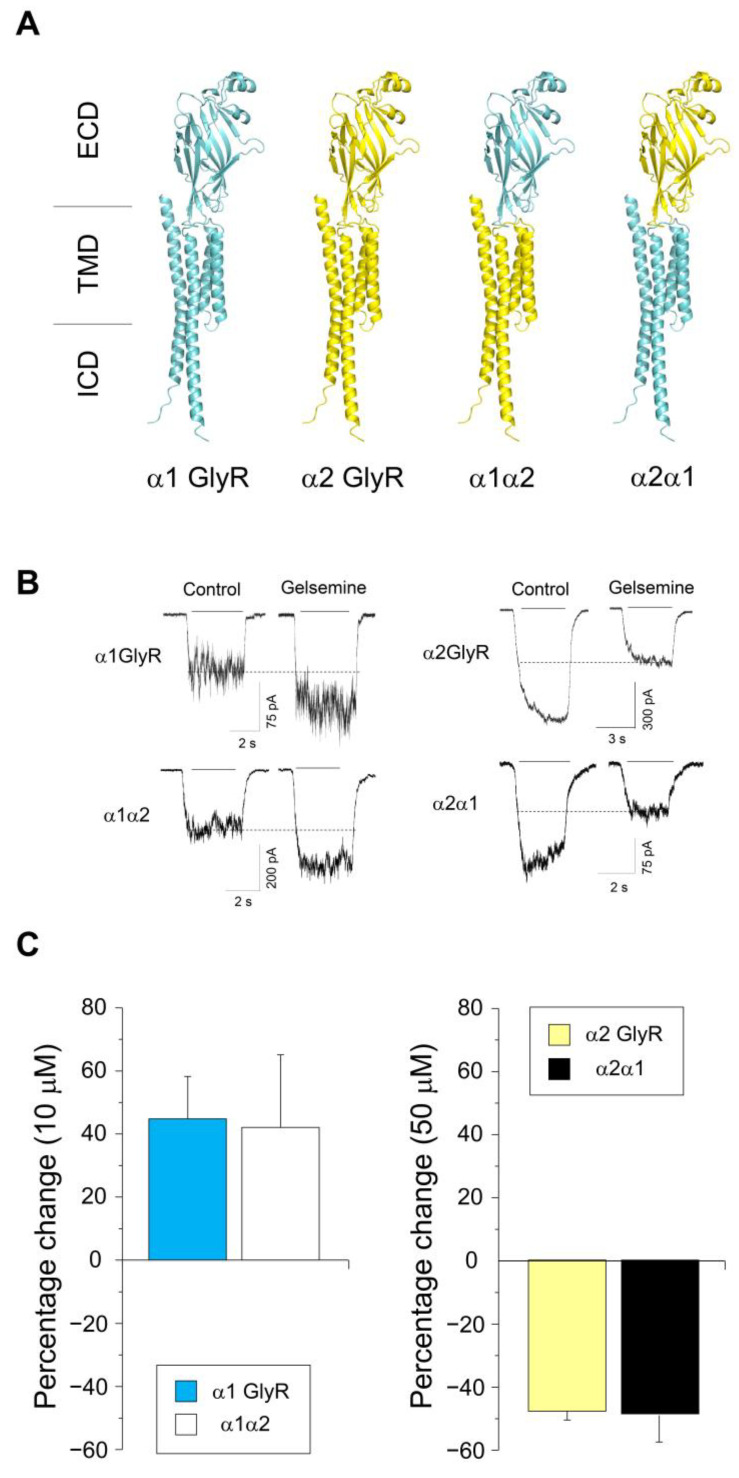
Relevance of the ECD for the subunit-specific actions of gelsemine on α1GlyRs. (**A**) Structural outlook of wild-type and chimeric receptors studied. (**B**) Current traces show the effects of gelsemine (10 μM or 50 μM) on wild-type and chimeric α1α2 or α2α1 GlyRs. Currents were evoked using 35 μM (α1), 60 μM (α2), 35 μM (α1α2), and 65 μM (α2α1) of glycine. (**C**) Summary of gelsemine effects on wild-type and chimeric GlyRs. Differences were not significant. α1 (n = 9), α2 (n = 6), α1α2 (n = 7), α2α1 (n = 10). Unpaired Student’s *t*-test: 10 μM, *p* = 0.91; 50 μM, *p* = 0.93. Data are presented as means ± SEM.

**Figure 4 ijms-25-03390-f004:**
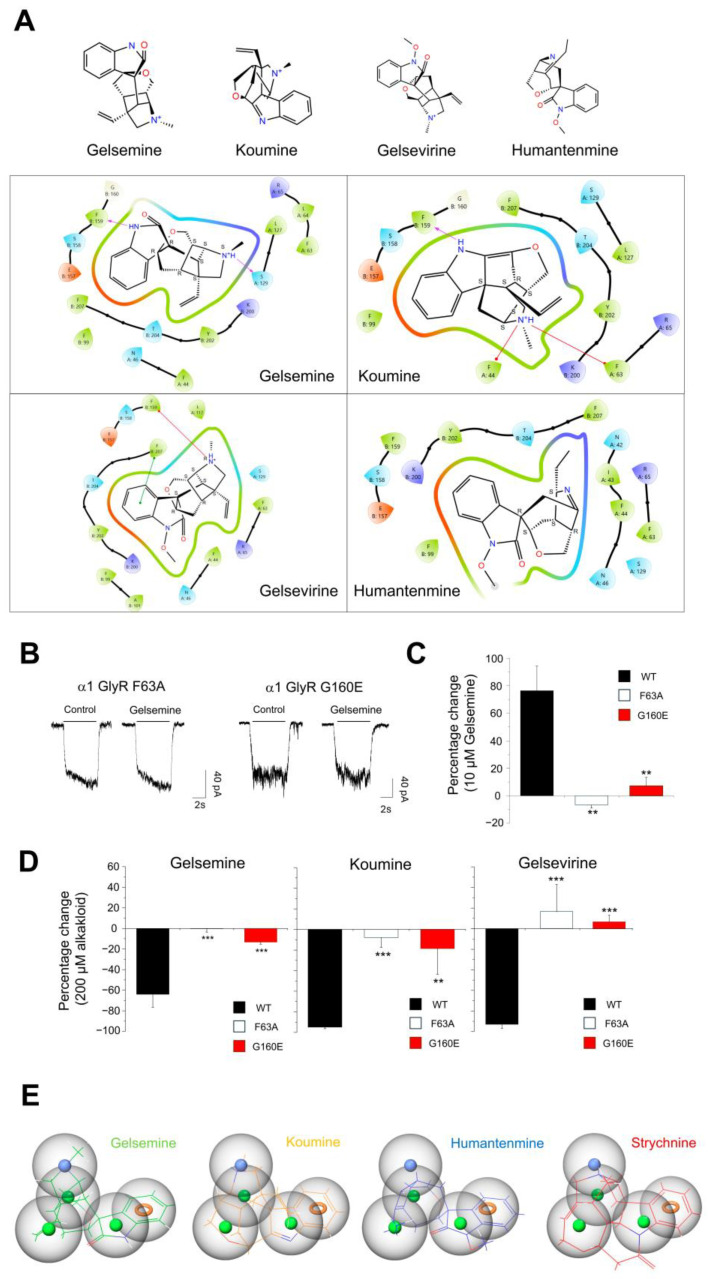
Molecular analysis of *Gelsemium* alkaloid’s interactions with amino acids within the orthosteric site of α1GlyRs. (**A**) Two-dimensional structures of gelsemine, koumine, gelsevirine, and humantenmine (pH 7.0) and interaction diagrams of the alkaloids with residues of the orthosteric site of α1GlyRs. Interactions between α1GlyR and each alkaloid are described (4 Å cutoff). The purple arrows indicate hydrogen bonds, while the red lines represent pi–cation interactions. The green line symbolizes a pi–pi interaction. Numbered residues are depicted by colored drops. The color code describes the amino acid properties (green, hydrophobic residues; red, negatively charged residues; blue, positively charged residues; cyan, polar residues; light yellow, glycine). (**B**) Sample current traces showing the sensitivity loss to gelsemine (200 μM) of α1GlyR F63A and G160E mutants. The currents were evoked using 2 mM (F63A) or 1 mM (G160E) of glycine. (**C**,**D**) The bar plots describe the potentiation percentage induced by gelsemine (**C**) or the inhibition percentage induced by gelsemine, koumine, or gelsevirine (**D**) on wild-type or mutated receptors. For graph (**C**), WT (n = 9), F63A (n = 4), G160E (n = 9). For graph (**D**), gelsemine, WT (n = 6), F63A (n = 4), G160E (n = 6); koumine, WT (n = 5), F63A (n = 4), G160E (n = 5); gelsevirine, WT (n = 6), F63A (n = 4), G160E (n = 6). ANOVA followed by Tukey post hoc test. Differences were significant. **, *p* < 0.01, F(2, 18) = 3.41, gelsemine-induced potentiation of wild-type α1GlyRs versus F63A and G160E (**C**). For gelsemine inhibition: ***, *p* < 0.001, F(2, 11) = 1.47; for koumine inhibition: **, *p* < 0.01, ***, *p* < 0.001, F(2, 10) = 2.50; for gelsevirine inhibition: ***, *p* < 0.001, F(2, 12) = 0.32 (**D**). Data are presented as means ± SEM. (**E**) Pharmacophore modeling of *Gelsemium* alkaloids. Strychnine is also shown as a reference competitive alkaloid.

**Table 1 ijms-25-03390-t001:** Pharmacological actions of koumine, gelsevirine, and humantenmine on GlyRs.

	Alkaloid	IC_50_ (μM)	n_H_	Maximal Modulation (%)	Maximal Modulation Concentration (μM)	K_i_ (μM) *	n
α1 GlyR	Koumine	31.5 ± 1.7	1.7 ± 0.1	−95.0 ± 1.5	300	53.1 ± 17.1	6
	Gelsevirine	40.6 ± 8.2	1.1 ± 0.1	−92.7 ± 4.1	300	50.1 ± 9.5	9
	Humantenmine	ND	ND	−19.2 ± 7.8	300	ND	6
α1β GlyR	Koumine	32.5 ± 13.2	0.9 ± 0.5	−90.8 ± 2.3	300	50.1 ± 9.5	4
	Gelsevirine	65.2 ± 6.5	3.5 ± 0.9	−88.6 ± 5.0	200	88.5 ± 13.6	4
	Humantenmine	ND	ND	−7.33 ± 10.8	0.01	ND	5
α2 GlyR	Koumine	11.2 ± 2.1	1.0 ± 0.2	−94.3.0 ± 2.7	200	44.2 ± 4.6	6
	Gelsevirine	40.1 ± 5.5	2.4 ± 0.7	−99.1 ± 0.3	300	73.9 ± 21.7	4
	Humantenmine	ND	ND	8.7 ± 9.6	50	ND	6
α2β GlyR	Koumine	23.6 ± 8.3	1.3 ± 0.6	−91.5.4 ± 5.4	300	57.4 ± 4.0	4
	Gelsevirine	38.9 ± 7.7	3.0 ± 1.5	−96.9 ± 0.5	300	25.2 ± 15.0	4
	Humantenmine	ND	ND	−10.3 ± 11.2	50	ND	4
α3 GlyR	Koumine	15.8 ± 4.6	0.9 ± 0.2	−87.3 ± 4.0	300	33.9 ± 15.8	8
	Gelsevirine	4.0 ± 0.7	0.7 ± 0.1	−97.9 ± 2.7	200	8.7 ± 5.4	9
	Humantenmine	ND	ND	−20.2 ± 10.7	0.1	ND	4
α3β GlyR	Koumine	20.2 ± 7.7	0.9 ± 0.3	−90.9 ± 3.15	300	54.1 ± 20.0	7
	Gelsevirine	14.0 ± 4.3	0.9 ± 0.2	−77.1 ± 4.8	300	30.1 ± 19.5	9
	Humantenmine	ND	ND	−15.6 ± 3.8	300	ND	3

* The K_i_ was calculated using the Cheng–Prussof equation. ND, not determined. Statistical comparisons between homomeric receptors (ANOVA followed by Tukey post hoc test). For koumine: IC_50_, F(2, 12) = 1.08. Significant difference (*p* < 0.01) between α1 and α2; K_i_, F(2, 12) = 2.40. Differences were not significant; n_H_, F(2, 12) = 3.82. Differences were not significant. For gelseverine: IC_50_, F(2, 17) = 3.58. Significant difference (*p* < 0.05) between α1 and α3; K_i_, F(2, 17) = 0.13. Significant difference (*p* < 0.05) between α2 and α3. n_H_, F(2, 17) = 9.99. Significant difference (*p* < 0.0001) between α2 and α3 GlyRs. Significant difference (*p* < 0.001) between α1 and α2 GlyRs. The percentages of maximal modulation of koumine and gelsevirine were different from control currents (paired *t*-Test, *p* < 0.001). The percentages of maximal modulation of humantenmine were not significantly different from control (paired *t*-Test: α1, *p* = 0.24; α2, *p* = 0.29; α3, *p* = 0.17). Statistical comparisons between heteromeric receptors (ANOVA followed by Tukey post hoc test). For koumine: IC_50_, F(2, 9) = 0.68. Differences were not significant; K_i_, F(2, 9) = 2.40. Differences were not significant; n_H_, F(2, 9) = 1.91. Differences were not significant. For gelseverine: IC_50_, F(2, 11) = 0.08. Significant difference (*p* < 0.01) between α1β and α3β; K_i_, F(2, 11) = 0.13. Significant difference (*p* < 0.05) between α1β and α3β; n_H_, F(2, 11) = 0.95. Differences were not significant. The percentages of maximal modulation of koumine and gelsevirine were different from control currents (paired *t*-Test, *p* < 0.001). The percentages of maximal modulation of humantenmine were not significantly different from control (paired *t*-Test: α1β, *p* = 0.63; α2β, *p* = 0.76; α3β, *p* = 0.20).

## Data Availability

The data used to support the findings of this study are already incorporatedwithin this manuscript.

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
