# Peer review of "Molecular Pharmacology of Gelsemium Alkaloids on Inhibitory Receptors"

_ijms, 2024, doi:10.3390/ijms25063390_

Round 1

Reviewer 1 Report

Comments and Suggestions for Authors

The manuscript by Marileo et al. describes the actions of several alkaloids on Gly and GABA-A receptors. Functional electrophysiology is accompanied by modeling data, together indicating that the compounds inhibit the receptors through interactions with the transmitter binding sites. Overall, the observations in this quite nice study are of interest to the scientific community. There are, however, a few issues that need to be addressed first.

1. Experimental part.  Please confirm the competitive inhibition mechanism by comparing IC50s (may pick any one compound) at low and high agonist concentrations. If the compound acts as competitive inhibitor at the transmitter binding site then its IC50 will be higher when measured at higher transmitter concentrations. Alternatively, and ideally, the paper would show Schild analysis.

2. Presentation.

Abstract: should it be bioactive rather than active? "direct modulator" may not be the best term, perhaps just modulator? Include a sentence on GABA-AR results.

Introduction: l. 49, should be "in models".  Same comment as above regarding "direct functional modulator". l. 66, surely excitatory ligand-gated ion channels like glutamate receptors also contribute to control excitability of the CNS.

Results: l. 82, are the IC50 values best-fit parameters ± SE of the fit from pooled data, or are they means and S.D. of fits to data from each individual cell separately? Clarify here or in Methods.  What were the concentrations of glycine and GABA throughout the study? Provide this, and the respective EC values (ideally, open probability values) in the text and/or figure legends. The docking scores for all compounds (Fig. 2B) in alpha1 look the same. Then why is there such a difference in functional effects (HUM vs. others)? For the data on chimeric receptors, provide glycine concentrations and EC values.  Modeling: I am unclear how is it that the rather large alkaloid molecules fit in the small binding site for glycine?  In Figures 2A and 4A, show how glycine binds and interacts.

Methods: provide species names for the receptor subunits.

Figures: the resolution in all figures is very poor, to the degree that one has to rely on text to learn what is there.  In Fig. 2B, D, what do the boxes, horizontal lines and whiskers correspond to? Clarify in figure legend.  Also, is lower (more negative) docking score "good" or "bad"? Add a sentence clarifying this in figure legend. 

3. General.  Is the Gelsemium plant used in traditional medicine? What is the rationale for these studies? Is it a purely academic study of compounds X-Z acting on the receptors that the lab happens to study? Some of this is provided in Discussion but I think these basic issues need to be addressed.

Comments on the Quality of English Language

Some editing is needed.

Author Response

The manuscript by Marileo et al. describes the actions of several alkaloids on Gly and GABA-A receptors. Functional electrophysiology is accompanied by modeling data, together indicating that the compounds inhibit the receptors through interactions with the transmitter binding sites. Overall, the observations in this quite nice study are of interest to the scientific community. There are, however, a few issues that need to be addressed first.

We are pleased to learn that the reviewer found our work interesting for the scientific community despite its limitations. We appreciate all the comments and questions, as we genuinely believe they have contributed to improving our manuscript. The revised manuscript has been reviewed by a native speaker. We apologize for any language-related mistakes present in the previous version.

Likewise, we feel relevant to inform that we request an extension of three weeks from the first deadline informed by the editorial office (16th February). This extension request is related to the closure of our university during the summer break, which encompasses most of February, as well as the complexity involved in performing additional electrophysiological recordings, which are time-consuming assays. Executing additional electrophysiological experiments is even more challenging during the summer break, primarily due to a lack of personnel. The editorial office has partially accepted our request for such an extension, further complicating the execution of additional assays. However, we believe that many of the concerns raised can be addressed through a more detailed discussion of the current literature, possibly complemented by additional electrophysiological experiments.

  1. Experimental part.  Please confirm the competitive inhibition mechanism by comparing IC50s (may pick any one compound) at low and high agonist concentrations. If the compound acts as competitive inhibitor at the transmitter binding site then its IC50 will be higher when measured at higher transmitter concentrations. Alternatively, and ideally, the paper would show Schild analysis.

Answer: 

We appreciate this important question and recognize the need for clarification in our manuscript.

The competitive nature of Gelsemium alkaloids on inhibitory channels has been investigated in previous studies by various research groups. One line of evidence comes from radio-ligand competitive displacement studies conducted by Wang’s group (Zhang et al., 2013; Shoaib et al., 2019). Using native glycine receptors from spinal cord tissue, these studies found that the alkaloids gelsemine and koumine displaced the binding curves of 3H strychnine to the right. For instance, Zhang and colleagues reported IC50 and Ki values of gelsemine on native glycine receptors of approximately 40 and 22 μM, respectively.

A second line of evidence comes from electrophysiological studies on recombinant and native glycine and GABA-A receptors (Lara et al., 2016; Marileo et al., 2023). These studies focused solely on gelsemine's modulation of both ion channels (no other Gelsemium alkaloids were tested). Lara and colleagues found that gelsemine displaced the glycine concentration response curves of recombinant alpha1, alpha2, and alpha3 glycine receptors, depending on their functional actions (i.e., displacement to the left in the case of alpha1, and to the right in the case of alpha2 and alpha3). The IC50 for gelsemine inhibition of the glycinergic current through spinal glycine receptors was approximately 42 μM. Similarly, for GABA-A receptors, Marileo and colleagues found that gelsemine displaced the GABA concentration response curve to the right, consistent with competitive inhibition.

Thus, we believe that a combined analysis of the radioligand assays and electrophysiological analyses discussed above supports the notion that gelsemine modulation of glycine and GABA-A receptors, at least in a significant part, is competitive in nature. Given their relevance to our work, the findings discussed above are now included in the updated manuscript (Introduction, lines 76-90). Since the functional actions of other Gelsemium alkaloids on inhibitory receptors have not yet been reported, we believe that additional experiments exploring and confirming the competitive nature of these interactions are valuable. We believe that the electrophysiological results and in silico data reported in the present manuscript provide further evidence supporting the interaction of additional Gelsemium alkaloids with the orthosteric site of GlyRs. In this regard, the electrophysiological experiments suggested by the reviewer are intriguing and hold high functional significance. However, as explained earlier, the editorial office did not provide sufficient time to conduct these assays. In the meantime, we are endeavoring to perform these experiments for further review of this article or for future publications. We hope the reviewer understands this challenging situation.

  1. Presentation.

Abstract: should it be bioactive rather than active? "direct modulator" may not be the best term, perhaps just modulator? Include a sentence on GABA-AR results.

Answer: The suggestions have been incorporated into the abstract, and a sentence on GABAA-Rs has been added. (Lines 19,27).

 Introduction: l. 49, should be "in models".  Same comment as above regarding "direct functional modulator". l. 66, surely excitatory ligand-gated ion channels like glutamate receptors also contribute to control excitability of the CNS.

Answer. The corrections have been incorporated, and we have edited the phrase regarding excitability control. Thank you. (lines 73-75).

Results: l. 82, are the IC50 values best-fit parameters ± SE of the fit from pooled data, or are they means and S.D. of fits to data from each individual cell separately? Clarify here or in Methods.  What were the concentrations of glycine and GABA throughout the study? Provide this, and the respective EC values (ideally, open probability values) in the text and/or figure legends.

Answers. The IC50 values correspond to best-fit mean ± SE from pooled data. This clarification has been included in the Method section. The concentrations of glycine and GABA used to study alkaloid sensitivity, as well as the EC50 values for glycine or GABA sensitivity of the receptor subtypes studied, have been included throughout the manuscript.  (lines 148, 151, 256, 326, 472-478, 486).

For the data on chimeric receptors, provide glycine concentrations and EC values.

Done (lines 256)  

Modeling: I am unclear how is it that the rather large alkaloid molecules fit in the small binding site for glycine?  In Figures 2A and 4A, show how glycine binds and interacts.

The docking scores for all compounds (Fig. 2B) in alpha1 look the same. Then why is there such a difference in functional effects (HUM vs. others)?

We appreciate this insightful question. Docking scores (DS) represent predicted binding affinities (in kcal/mol) related to the ability of each molecule to interact with a defined binding site. These in silico assays calculate feasible ligand-protein interactions within a protein structure at a given site, but they cannot predict potential effects after binding. In the assays described (Figure 2), calculations were confined to the orthosteric site of the receptors. For each ligand-receptor interaction, referred to as a binding pose, a docking score is computed. Consequently, various alkaloids may yield diverse docking score values when binding to the receptor's molecular site, resulting in multiple binding poses.

Traditionally, DS datasets are interpreted by comparing the 'best' docking score absolute values between conditions, typically represented by the most negative value, indicative of a lower energy and a more stable ligand-receptor complex. However, we argue that this approach overlooks a significant portion of potential ligand-receptor interactions. Therefore, we propose using box representations (percentile 25 and 75, along with median) in conjunction with maximal and minimal docking score values (indicated by whiskers) for a comprehensive description of the complete DS dataset. Notably, while the alkaloid humantenmine may interact with the orthosteric site in silico, its docking score distribution tends towards less negative values, approaching zero. In contrast, strychnine and bicuculline, known reference competitive antagonists of glycine and GABAA receptors respectively, exhibit distributions skewed towards more negative values.

A part of these explanations has been incorporated into the text and the legend of Figure 2 (lines 191-199, 232-237). Overall, we believe that the DS dataset (Figure 2) offers structural insights into the interactions of Gelsemium alkaloids with the orthosteric sites of glycine and GABAA receptors. However, it's important to note that DS calculations should not be interpreted as a direct measure of functional modulation. Instead, they serve as a theoretical qualitative description of the energetic profile of alkaloid binding to a specific receptor site. These profiles are partly associated with, but cannot predict, the potential functional modulation of the channel by a given alkaloid.

Modeling: I am unclear how is it that the rather large alkaloid molecules fit in the small binding site for glycine?  In Figures 2A and 4A, show how glycine binds and interacts.

To comprehend how large alkaloid molecules fit into the small glycine-binding site of glycine receptors, it's crucial to consider the structural aspects and binding mechanisms involved. Analyzing the binding of strychnine, the classical glycine receptor antagonist, provides insight as it competes with glycine for binding at the extracellular inter-subunit sites of the receptor complex. Recent high-resolution structures of homo and heteromeric glycine receptors have confirmed the presence of orthosteric binding sites targeted by strychnine. Binding determinants for strychnine, such as Thr220 and Tyr218, partially overlap with those of glycine, indicating a shared binding region (Du et al. 2015; Gibbs et al. 2023; Yu et al. 2021; Huang et al. 2015). Strychnine, as the reference competitive antagonist of glycine receptors, maintains the receptor in its closed conformation, with a cavity in the orthosteric binding site larger compared to the open state, facilitating the binding of larger molecules such as alkaloids. In comparison, glycine utilizes only a small portion of the orthosteric site's total space, yet both are capable of binding to it. Glycine binding maintains the receptor in the open state.

As suggested by the reviewer, we have included glycine binding to the orthosteric site in the manuscript. However, as explained above, the glycine binding to the orthosteric site generates an open conformation of the channel, while strychnine (and the Gelsemium alkaloids) produces channels in the closed state. This critical difference complicated the integration of the glycine binding information into Figures 2 and 4, where all the alkaloids are shown using closed state channels. We believe that the structural features of alpha1 GlyRs bound to glycine, in the open state, are better described as an individual Supplementary Figure. Therefore, we have added this Figure (Figure S3, line 226) to enhance the clarity of our study. We hope the reviewer understands this modification.

Methods: provide species names for the receptor subunits.

Answer: Done (lines 442-445).

Figures: the resolution in all figures is very poor, to the degree that one has to rely on text to learn what is there.  In Fig. 2B, D, what do the boxes, horizontal lines and whiskers correspond to? Clarify in figure legend.  Also, is lower (more negative) docking score "good" or "bad"? Add a sentence clarifying this in figure legend. 

Answer. We apologize for the irregular figure resolution. In the revised manuscript, we have provided figures with better resolution. Additionally, we have expanded the description of the boxes, horizontal lines, and whiskers of the graphs in the Figure 2 legend. Furthermore, a more comprehensive explanation of the docking score concept and the interpretation of lower or higher values has been included in the text (lines 191-199, 232-237).

  1. General.  Is the Gelsemium plant used in traditional medicine? What is the rationale for these studies? Is it a purely academic study of compounds X-Z acting on the receptors that the lab happens to study? Some of this is provided in Discussion but I think these basic issues need to be addressed.

Answer. Thank you for this insightful question. We have expanded the description of the ethnopharmacological relevance of Gelsemium plants in the introduction. Additionally, we have included a more detailed explanation about the rationale guiding these studies (lines 41-50, 94-98).

Reviewer 2 Report

Comments and Suggestions for Authors

In this work the authors test the major indole alkaloids from Gelsemium  on GABA and GlyR response, and use bioinformatic approaches to probe possible interactions at the orthosteric binding site.  Some of these are tested with 2 mutant receptors. However the data from these mutant receptors does not seem compatible with previously published data on the same mutants, which does need some explanation.  The low potency of the alkaloids is also not really considered i.e effects at the receptors in vivo is unlikely.  Nevertheless  this is  an interesting study with a nice range of data, although the writing and especially the english needs improving in a number of places ( some of which are detailed below).  

Abstract

L 23  remove ‘aim to’  - this is not the introduction

L24  Remove ‘the for major indole alkaloids ( i.e. 

Replace ‘with ‘ with ‘and’

L25  potency gives no information  here. Replace with IC50 values

L27 Remove ‘electrophysiological and bioinformatic assays” – these are mentioned above

 L32 Either contributing to or expanding but not both

INtroduction

L63  ( also L98 L110) Majorly is a poor English. Reword or remove

L66 What does gelsemine do?  ( inhibit/enhance, concn etc?)

L70 ‘…whether indole alkaloids other than gelsemine exert …’ is better english

Results

L84 A change that is not statistically significant should not be reported as a ‘small change’

L104  - sentence is incomplete .  Add units

Figure 1 A C E  what conc of glycine was used here?

Table 1”; Use Ki or Kb but not Kd. Add details of the cheng prussof equation and how the epxts were done.  Indicate which differences are statistically different to control.  Why are the Kd and IC50 identical at a1GlyR for gelsevirine but not elsewhere, and why is the Kd  lower than the IC50 at a2b R?  .   

Explanation needed somewhere in  the text: Why is the nH much higher for gelsevirine in many Rs?  Why is the max concn used for humantenmine much lower in most receptors? Could this be why is has no inhibition?

L132  Were the structures in the open or closed configuration?

L146 by values do the authors mean profiles? Not clear

L148  Fig 2B does not show zero scores for humantenmine – its values overlap with all the others. Clarify.

L159  & L187 replace ‘discarded’ – poor english

Fig2A&C How many docked poses were obtained and how were the ones shown chosen?  Larger images of the docked compounds and important residues labelled would be good.

Fig2B &D Negative controls needed here  e.g bic and stn

Fig3  N= ?

L204  Could this have been a cation pi interaction?  Needs discussing

L217 Mutations of many of these residues has been reported by several groups  e.g. Schofield, Betz, Vandenberg, Tang etc .  Some have dramatic effects on binding e.g. F63A increased gly EC50 by >1000 fold ( If the alkaloid EC50s increased by the same amount it is not surprising they have no effect). How do the authors explain that their responses in this mutant look similar to WT ?

Fig 4 A the figures in the boxes are very difficult to understand  -  I suggest removing the green and blue blobs ( what are they?) and showing and labelling the binding site residues in particular the atoms that potentially interact with the compounds

B  what concn of glycine was used here?  Add a positive and negative control e.g. stn and bic

Explain the colored blobs

Discusssion

It is not clear to me if the authors think their data explains the toxicity of the alkaloids or does not, as they argue both for and against. They need to come down on one side or the other but also explain the other point of view.   They also need to provide an approximate concentration that needs to be ingested for toxicity and explain if it is consistent with an  IC50 of 10s of micromolar.  

Some English correction  needed here  e.g  L274 ‘worth to describe’ is poor

L 297  Add have

L299  What are the  beneficial effects ? Not mentioned before.

L320  to not with

L324 In not on

L329 replace it will be possible to  with ‘we’

Materials and methods

Were these all human Rs?

What are their Uniprot numbers?

L369 on daily bases  - doesn’t make  sense

Molecular docking

Insufficient details given here for anyone to repeat this. Much more information needed.

Comments on the Quality of English Language

poor in places

Author Response

In this work the authors test the major indole alkaloids from Gelsemium on GABA and GlyR response, and use bioinformatic approaches to probe possible interactions at the orthosteric binding site.  Some of these are tested with 2 mutant receptors. However the data from these mutant receptors does not seem compatible with previously published data on the same mutants, which does need some explanation.  The low potency of the alkaloids is also not really considered i.e effects at the receptors in vivo is unlikely. Nevertheless  this is  an interesting study with a nice range of data, although the writing and especially the english needs improving in a number of places ( some of which are detailed below).  

We are pleased to hear that the reviewer found our work interesting, despite its limitations. We appreciate all the comments and questions, as we truly believe they have contributed to improving our manuscript. The revised manuscript was reviewed by a native speaker. We apologize for any language-related mistakes in the previous version.

Abstract

L 23  remove ‘aim to’  - this is not the introduction

Answer. We rephrase the sentence.

L24  Remove ‘the for major indole alkaloids ( i.e.  ‘

Replace ‘with ‘with ‘and’

Answer. Done.

L25  potency gives no information  here. Replace with IC50 values

Answer: The sentence has been rephrased, and several IC50 values have been added. However, we did not include additional IC50 values due to limitations on abstract word count. (Line 25).

L27 Remove ‘electrophysiological and bioinformatic assays” – these are mentioned above

Answer: Done

 L32 Either contributing to or expanding but not both

Answer: We rephrase the sentence Lines 32-33

INtroduction

L63  ( also L98 L110) Majorly is a poor English. Reword or remove

Answer. We rephrase the sentences. Thank you.

L66 What does gelsemine do?  ( inhibit/enhance, concn etc?)

Answer. A more comprehensive description of gelsemine's effects on glycine and GABAA receptors has been added (Lines 76-90)

L70 ‘…whether indole alkaloids other than gelsemine exert …’ is better English

Answer. Many thanks for your suggestion. The text was modified accordingly. Lines 94-95.

Results

L84 A change that is not statistically significant should not be reported as a ‘small change’

Answer. We edited the phrase accordingly. Line 111.

L104 - sentence is incomplete .  Add units

Answer. Done. Thank you. Line 129.

Figure 1 A C E  what conc of glycine was used here?

Answer. The glycine and GABA concentrations have been included in the figure legends (lines 148, 151, 256, 326).

Table 1”; Use Ki or Kb but not Kd. Add details of the cheng prussof equation and how the epxts were done.  Indicate which differences are statistically different to control.  Why are the Kd and IC50 identical at a1GlyR for gelsevirine but not elsewhere, and why is the Kd lower than the IC50 at a2b R? 

Use Ki or Kb but not Kd. Add details of the cheng prussof equation and how the epxts were done. 

We have included the methodological details and the Cheng Prussof equation in the Materials and Methods section (lines xxx). Additionally, we now refer to the calculation as "Ki" (lines 489-492).

Indicate which differences are statistically different to control.

We have included a statistical comparison of the sensitivity to the alkaloids between homomeric and heteromeric receptor subtypes (lines 161-173).

Why are the Kd and IC50 identical at a1GlyR for gelsevirine but not elsewhere, and why is the Kd lower than the IC50 at a2b R? 

The equal values of the Kd and IC50 of a1GlyR for gelsevirine correspond to a typographic error, and the value is now corrected. In the case of a2b, the average Kd of gelsevirine is indeed lower than the corresponding IC50. However, the Kb values for gelsevirine on a2b compared with a1b or a3b receptors were not significantly different (Lines 107-109, 118-123).

Explanation needed somewhere in the text: Why is the nH much higher for gelsevirine in many Rs?  

Answer. We appreciate the reviewer's observation. Regarding the modulation by gelsevirine, we found significant differences in nH among homomeric receptors. Specifically, significant differences were observed between α2 and α3 GlyRs, as well as between α1 and α2 GlyRs (Lines 107-109, 118-123). Interestingly, heteromeric GlyRs did not exhibit significant differences, suggesting that the incorporation of the b subunits into the receptor complex may somehow regulate the cooperativity of the alkaloid binding sites. This idea is supported by the molecular modeling data on heteromeric GlyRs, which show interactions between the alkaloids and the α-β binding interphase (Figure S2). On the other hand, the nH for koumine generally displayed lower values, suggesting that the higher nH observed for gelseverine is, at least in part, specific to this alkaloid. We have incorporated some of these comments into the Results section (Lines 107-109, 118-123).

Why is the max concn used for humantenmine much lower in most receptors? Could this be why is has no inhibition?

Answer. The sensitivity of humantenmine was assessed using concentrations ranging from 0.01 to 300 μM in all recorded cells (see Figure 1B-D). In some cases, the maximal modulation of the glycine current by humantenmine was not always coincident with the higher alkaloid concentrations (200 or 300 μM) but was reached with lower concentrations. The maximal percentages of change observed with humantenmine were not significantly different from the control (Lines 166-167, 172-173).

L132  Were the structures in the open or closed configuration?

Answer. All the structures utilized bound either strychnine (for glycine receptors) or bicuculline (for GABAA receptors), indicating that they are in a closed conformational state (line 496).

L146 by values do the authors mean profiles? Not clear

Answer. We have replaced the word 'profile' with 'values' since our intention is to demonstrate that the docking score values, and the amino acids involved in the interactions, were similar between the Gelsemium alkaloids and strychnine.

L148  Fig 2B does not show zero scores for humantenmine – its values overlap with all the others. Clarify.

Answer. Our calculations suggest that humantenmine can interact with glycine receptors, albeit likely with lower binding affinities compared to other alkaloids. This suggests that its docking score values are higher than those of gelsemine, although they do not reach zero (or positive values), which would indicate a completely unfavorable interaction (lines 191-199).

L159  & L187 replace ‘discarded’ – poor English

Answer. We edited the sentence.

Fig2A&C How many docked poses were obtained and how were the ones shown chosen?  Larger images of the docked compounds and important residues labelled would be good.

Answer. To obtain the various binding poses for each alkaloid, we utilized the conformations generated by LigPrep for gelsemine (13), koumine (30), gelseverine (19), hummantenmine (62), strychnine (37), and bicuculline (16). This information has been incorporated into the legend of Figure 2 (lines 235-237). The images have been updated to enhance their resolution. For the figures, we have presented the complexes with the lowest docking score (most favorable energy).

Fig 2B &D Negative controls needed here  e.g bic and stn

Answer. Strychnine and bicuculline are represented in the docking score graph for glycine and GABAA receptors, respectively (Figure 2B-D)

Fig3  N= ?

Answer. We have incorporated the number of cells in the legend of Figure 3 (lines 258-259). Thank you for this suggestion.

L204  Could this have been a cation pi interaction?  Needs discussing

Answer. Indeed, the presence of cation pi interaction is observed in gelseverine and koumine (Fig. 4), and given the chemical properties of the alkaloids, such interactions would be anticipated. We have incorporated this information into the results text and in the figure 4 legend (Lines 270-275).

L217 Mutations of many of these residues has been reported by several groups  e.g. Schofield, Betz, Vandenberg, Tang etc .  Some have dramatic effects on binding e.g. F63A increased gly EC50 by >1000 fold ( If the alkaloid EC50s increased by the same amount it is not surprising they have no effect). How do the authors explain that their responses in this mutant look similar to WT ?

We appreciate the reviewer's insightful question. As correctly explained by the reviewer, the F63A mutant exhibits impaired glycine sensitivity. However, in HEK293 cells expressing this mutant, we were able to record small currents (approximately 100 pA) at sub-saturating glycine concentrations with sufficient stability to conduct the analysis of Gelsemium alkaloid sensitivity. To perform these experiments, the currents were elicited using 2 mM glycine, and the recordings were conducted after a longer post-transfection period (2-3 days). Towards the end of these recordings, a higher concentration of glycine (10 mM) was applied to confirm that the working glycine concentration corresponded to a sub-saturating level. Concentrations higher than 10 mM of glycine (e.g., 15-20 mM) were difficult to test due to seal variabilities. From our recordings, we estimated an EC50 of at least 8 mM. Additionally, using 2 mM glycine, we assessed the strychnine sensitivity (2 μM) of the F63A mutant and found no inhibition, which aligns with previous electrophysiological findings (Grudzinska et al., 2005). In contrast, the G160E mutation exhibited significantly diminished sensitivity to Gelsemium alkaloids but retained a significant portion of the sensitivity to strychnine (2 μM). A portion of this information has been incorporated into both the Methods and Results sections (lines 286-300, 476-479).

Fig 4 A the figures in the boxes are very difficult to understand  -  I suggest removing the green and blue blobs ( what are they?) and showing and labelling the binding site residues in particular the atoms that potentially interact with the compounds

Answer. We have rectified the issue arising from the low resolution of the figure. The blobs now serve as schematic representations of the amino acids surrounding the compounds. An explanation regarding the color of each amino acid, based on its properties, has been included in the Figure 4 legend (Lines 320-324).

B  what concn of glycine was used here?  Add a positive and negative control e.g. stn and bic

Answer. The glycine concentrations are included. Strychinne controls are described in the text. (lines 294-300, 326-327)

Discusssion

It is not clear to me if the authors think their data explains the toxicity of the alkaloids or does not, as they argue both for and against. They need to come down on one side or the other but also explain the other point of view.   They also need to provide an approximate concentration that needs to be ingested for toxicity and explain if it is consistent with an IC50 of 10s of micromolar.  

Answer. Previous electrophysiological results and the data from this manuscript indicate that the concentrations of alkaloids causing approximately a 25-50% decrease in currents through GlyRs and GABAARs are around 50 µM, which is approximately equivalent to 16 µg/ml. The maximal modulation concentrations, which typically result in a decrease of 80-95% of the currents, are reached with 100-300 µM, approximately equivalent to 30-95 µg/ml (Lara et al., 2016; Marileo et al., 2023; Table 1). These concentration ranges correlate well with lethal plasma concentrations of Gelsemium alkaloids reported in humans, which are in the range of 25-50 µg/ml for koumine and 13-30 µg/ml for gelsemine (Yang et al., 2022). Additionally, it should be noted that pharmacokinetic studies performed with 11 Gelsemium alkaloids described that gelsemine was the only compound detectable in brain tissue after 3 days of systemic application, while koumine was fully depleted after 1 day (Wu et al., 2022). Considering these observations, we believe that the functional inhibition exerted by gelsemine, koumine, and gelsevirine on GlyRs and GABAARs contributes, at least in part, to the toxic effects of these alkaloids. Nevertheless, our data with humantenmine, the most toxic Gelsemium alkaloid in terms of its LD50, suggest that there are other molecular targets involved in the toxic and lethal actions of Gelsemium alkaloids. These clarifications have been incorporated into the discussion (lines 407-419).

Some English correction needed here  e.g  L274 ‘worth to describe’ is poor

Answer. We rephrase the sentence. Thank you.

L 297  Add have

Answer. We edited the sentence.

L299  What are the  beneficial effects ? Not mentioned before.

Answer. The beneficial actions of gelsemine and koumine were described in the introduction section. (lines 39-50, 56-60)

L320  to not with

Answer. Done. Thank you.

L324 In not on

Answer. Done. Thank you.

L329 replace it will be possible to with ‘we’

Answer. We edited the sentence. Thank you.

 Materials and methods

Were these all human Rs?

Answer. The receptors used in this study are from rats. (lines 442-445).

What are their Uniprot numbers?

Answer. Uniprot numbers are provided in the methods section (lines 442-445).

L369 on daily bases  - doesn’t make  sense

Answer. We edited the sentence.

Molecular docking

Insufficient details given here for anyone to repeat this. Much more information needed.

Answer. We have expanded the information regarding the procedures involved in molecular docking. We appreciate this feedback. (Lines 494-517)

Round 2

Reviewer 1 Report

Comments and Suggestions for Authors

The authors have addressed my comments and criticism.

Comments on the Quality of English Language

N/A